# Energy Basics of Catalytic Hydrodesulfurization of Diesel Fuels

**Daria Petrova** [1,2,*] , **Valentina Lyubimenko** [1] , **Evgenii Ivanov** [1] , **Pavel Gushchin** [1,3,*] and **Ivan Kolesnikov** [1]

1   Department of Physical and Colloid Chemistry, Gubkin University, 119991 Moscow, Russia
2   Vladimir Zelman Center for Neurobiology and Brain Rehabilitation, Skolkovo Institute of Science and Technology, 121205 Moscow, Russia
3   Institute for Problems in Mechanical Engineering of the Russian Academy of Sciences (IPME RAS), 199178 St. Petersburg, Russia
*   Correspondence: petrova.msu@gmail.com (D.P.); guschin.p@mail.ru (P.G.); Tel./Fax: +7-4995078241 (P.G.)

**Abstract:** Currently, Euro 5 (no more than 10 ppm sulfur content) and Euro 6 (less than 10 ppm sulfur content) diesel motor fuels are produced worldwide. High-quality diesel fuels are produced by removing sulfur compounds using a hydrodesulfurization process. This article is devoted to the study of hydrodesulfurization of diesel fuel containing 120 ppm of sulfur compounds in the presence of an Al-Ni-Mo-O catalyst with a 98% diesel fuel purification rate. According to the Langmuir–Hinshelwood–Panchenkov theory, a kinetic model of the process is developed with the calculation of the theoretical change in the entropy and enthalpy of the activation of the hydrodesulfurization process. The mathematical model, for the first time, takes into account the influence of the pressure of substances involved in the process on the transformation of sulfur-containing compounds. A mechanism for diesel fuel hydrotreating from sulfur-containing compounds is formulated using a generalized quantum-chemical principle. The hydrodesulfurization mechanism includes nine stages. The formulated mechanism and developed mathematical model of hydrotreating fully describe the reaction of the hydrodesulfurization of diesel fuel and show the possibility of regulating and controlling this industrially important process.

**Keywords:** hydrotreatment; catalyst; kinetics; desulfurization; environmental standard

## 1. Introduction

The quality of diesel fuels on an industrial scale is improved at high-power hydrodesulfurization plants. The hydrodesulfurization process is carried out on solid catalysts of different compositions and properties. Aluminum-cobalt-molybdenum catalysts and catalysts based on such metals are most widely used in the industry. For scientific and practical purposes, the chemical composition, structure, state, and activity of such catalysts in hydrodesulfurization reactions of oil fractions are widely studied, highlighting the structural features of active centers. In a paper [1] on the catalyst Co-Mo-S, Mo-edge (Mo-E), Co-mixed Mo-edge (Mo-X), Co-edge (Co-E), and corner (Corner) sites were used to examine the structural effect of the CoMoS nanocluster on its hydrodesulfurization reaction activity. Short and long Mo-S bonds are identified where hydrogen dissociation occurs with an endo or exothermic effect and different activation energies. Co-Mo-S nanoclusters form coordination unsaturated centers that are highly active in the hydrogenolysis of the C-S bonds. At the corner centers, thiophene is desulfurized to produce 2-butene molecules with a selectivity of up to 95%. At higher temperatures, the lattice centers catalyze the synthesis of butadiene with a selectivity of up to 70% [2].

Iridium catalysts using different zirconium modified-SBA-15 supports were tested in the hydrotreating of tetralin and typical sulfur and nitrogen compounds present in diesel fuel [3]. It was observed that the presence of $Zr^{+4}$ had a remarkable effect on the dispersion and reducibility capacity of the iridium actives species. The inhibition effect in the hydrogenation of tetralin over

Ir-Zr-SBA15-ALD was: DBT < indole < quinoline < 4,6-DMDBT. However, high conversion of tetralin was achieved even when 300 ppm of S or N was added to diesel fuel [3].

The high catalytic activity of the catalyst is determined by the presence of a zirconium ion in its lattice as part of a tetrahedron and the presence of a mesoporous structure [4]. It was found that NiMo/Al$_2$O$_3$-ZrT-100 with a high Si/Zr ratio showed the highest HDS efficiency, which was 99.3%. The best operating conditions of the NiMo/Al$_2$O$_3$-ZrT-100 catalyst were obtained: a temperature of 350–360 °C, a pressure of 6 MPa, a WHSV of 1.0–1.2 h$^{-1}$, and the H$_2$ to oil ratio of 600 mL/mL.

To increase selectivity, the catalysts were loaded into the reactor in two layers [3,5]. A normal catalyst was loaded on top, and a catalyst containing promoters was placed in the lower layer. The selectivity of the process of hydrodesulfurization of sulfur compounds is significantly higher compared to the process of hydrogenolysis. The kinetic law of hydrodesulphation and hydrodenitrogenation of diesel fuel (DF), as well as hydrogenation of aromatic hydrocarbons, was studied in [6] in the presence of a NiMo/Al$_2$O$_3$ catalyst.

Thermodynamic analysis of diesel fuel hydrotreating reactions -hydrodesulfurization (HDS), hydrodenitrogenation (HDN) and hydrodearomatization (had)—was carried out by Gibbs free energy minimization [7]. Dibenzothiophene, naphthalene and pyridine were considered as representative compounds to study the HDhadHDA and HDN reactions, respectively [7]. It was found that the HDN reaction has no thermodynamic limitation, while HDA reaction has significant thermodynamic limitation at low pressures and high temperatures. Overall, HDS reaction has no thermodynamic limitation [7].

Works [8,9] are devoted to the discussion of the mechanisms of hydrodesulfurization of sulfur compounds. A study on the mechanism for thiophene hydrodesulfurization over zeolite L-supported sulfided Co-Mo catalysts using the density functional theory is presented in article [8]. It established that the pore framework of zeolite L plays a key role in decreasing the energy barrier by the stabilization effect [8]. The authors of paper [9] have presented a review of organic additives for hydrotreating catalysts and mechanisms of their action. Organic additives have a strong impact either on the metal ions oxidic state or on the activation process [9]. These organic molecules permit the maximization of the use of Co and Mo to generate an optimum content of the Co-Mo-S phase [9].

Work [10] aimed at selecting suitable support for a Co-Mo-Re catalyst for the hydrodesulfurization (HDS) of thiophenes. γ-Al$_2$O$_3$ and γ-Al$_2$O$_3$-HMS were used as a support for the Co-Mo-Re catalyst of HDS. It is shown that the catalyst supported on γ-Al$_2$O$_3$ displayed higher activity than the catalyst supported on γ-Al$_2$O$_3$-HMS [10]. The results led the authors to the conclusion that activity is favored by the suitable textural and acidic properties of the γ-Al$_2$O$_3$ support [10].

In paper [11], a hybrid model for HDS is proposed and validated using a lab-scale reactor and industrial data. The proposed hybrid structure is a combination of the mathematical model, optimization algorithm (offline mode), and Support Vector Regression data-driven model (on-line mode). The hybrid model can predict online product sulfur concentration with an error of 5–10%, thus giving a better opportunity to control the desulfurization process [11].

The morphology, microstructure, coordinative unsaturation, and hydrogenation activity of unsupported MoS$_2$ are studied in article [12]. It is found that coordinative unsaturation decreases with the growing temperature of reductive activation $T_R$. When $T_R$ was increased from low levels, activity in the ethene hydrogenation and chemisorption capacities of CO, O$_2$, and N$_2$O peaked, but at significantly different temperatures for activity and chemisorption. It was concluded that the drastic loss of chemisorption capacity after reduction at 873 K results predominantly from anisotropic MoS$_2$ crystallites preferentially exposing their basal planes at the surface of the polycrystalline particles [12].

It is shown in paper [13] that Ni atoms modify the structure and the orbital properties of the coordinatively unsaturated Ni-Mo-S active sites in the catalyst such that the hydrogenation reactants are easier to adsorb.

The catalyst $CoMoS/Al_2O_3$ used in study [14] was prepared by incipient wetness impregnation of $\gamma$-$Al_2O_3$ with a mixed solution contented elements of Co and Mo. The content of CoO and $MoO_3$ was 4.0 wt% and 8.0 wt%, respectively.

Depending on the process of their production, diesel fuels contain four classes of hydrocarbons: paraffins, olefins, alkylaromatics, and alkyl naphthenes, as well as various types of sulfur and nitrogen compounds [15,16]. The removal of sulfur and organometallic compounds by various industrial methods improves the quality and environmental friendliness of diesel fuel. The most common diesel fuel with a high environmental standard is Euro 5, which contains no more than 10 ppm of sulfur. According to the Euro 6 standard, diesel fuel must contain less than 10 ppm of sulfur.

It should be noted that, for the process of diesel fuel hydrotreating, oxygen-containing catalysts of different compositions and different nature are used: Co-Mo-O, W-Mo-Si-O, W-Mo-Al-O, $WS_2$, Ni-Mo-O, Ni-Co-O, Al-Co-Mo-O, Al-Ni-Mo-O, Ni-W-O, Co-W-O, as well as aluminosilicate catalysts promoted by oxides of Mo, Co, Ni, Ce, and zeolites [17–26]. Nickel-based catalysts [18,22,27,28] are the most widely used in the process of HDS of diesel fuel, while $MoS_2$-based catalysts [19,29–31] and CoMo/$\gamma$-Al2O3 [32,33] are less common. However, in most cases, the influence of various parameters on the degree of conversion of sulfur and nitrogen compounds has not been studied in detail.

The Mo-Ni/$\gamma$-$Al_2O_3$ nanoporous catalyst showed an increased activity in the decomposition of sulfur compounds, equal to 99 wt%, compared to the catalyst with the usual pore distribution [29,31]. However, the mechanism of the hydrodesulfurization process from the kinetic and quantum-chemical point of view has not been modeled.

Nanostructured catalysts $XMo_6$ (S)/$\gamma$-$Al_2O_3$, where $X$ = Al, Ga, In, Fe, Co, Ni based on aluminum nanoxide [20] were active in the conversion of sulfur-containing compounds in the range of 99–99.9%. Catalysts of this composition show increased activity in the desulfurization of diesel fuel from thiophene to dibenzothiophene, with the highest activity being In- and Ni-based catalysts.

The authors of [33] derived a pseudo-first-order kinetics equation when studying the desulfurization of dibenzothiophene in the presence of a Ni-Mo/$\gamma$-$Al_2O_3$ catalyst at 260 °C. The conversion rate of dimethyl dibenzothiophene ranged from 66 to 92%. Also, the kinetics in the presence of the Ni-Mo/$\gamma$-$Al_2O_3$ catalyst was studied for the oxidative desulfurization of dibenzothiophene in a solution of paraffin hydrocarbons at temperatures of 80–140 °C, and the first-order kinetics equation was derived [34]. The conversion of dimethyl dibenzothiophene ranged from 73 to 94%.

A two-stage process of desulphurization of the diesel fraction and model mixtures in the presence of Co-Mo/$Al_2O_3$, Ni-Mo/$Al_2O_3$ catalysts was used to achieve a residual sulfur content in the 50 ppm to 10 ppm range [35–37]. Based on the study of the kinetics of the diesel fraction desulfurization process, a pseudo-first-order equation was obtained [15].

It was found that the Ni-Mo/$Al_2O_3$ catalyst is more active in the hydrogenation process, and Co-Mo/$Al_2O_3$ is more active in the hydrodesulfurization process [23]. This is determined by the different degree of dispersion of the active phases on the catalyst surface and their selectivity, as well as the ratio of ensembles of polyhedra of the type {$CoO_4$·$MoO_6$}, {$MoO_4$·$MoO_6$}, {$NiO_4$·$MoO_4$}.

Review [22] presents hydrodesulfurization technologies for oil and petroleum products, in particular, a technology that combines adsorption pretreatment units for oil, diesel fuel, gasoline, and jet fuel with a conventional catalytic hydrodesulfurization unit for sulfur compounds.

Paper [24] presents a comprehensive review of recent developments in the field of aluminum oxide-based hydrodesulfurization catalysts for the production of sulfur-free petroleum products in particular diesel fuels. In this paper, the effect of various additions (chelating agents, new metals) in the hydrodesulfurization catalyst during synthesis, production of hybrid support and adaption of novel synthesis processes (like ultrasonic spray pyrolysis (USP), temperature-sensitive hydrothermal synthesis, rehydrationdehydration of alumina, monolith wash-coating, and EISA synthesis method, etc.) has been discussed.

This article is devoted to the study of thermodynamics, kinetics, and mechanisms of hydrodesulfurization of diesel fuels and the creation of kinetic and parametric models of diesel fuel purification from sulfur compounds as well as hydrocatalytic processes, taking into account thermodynamic equilibrium, the generalized quantum-chemical principle, the basic principle of chemical kinetics, and the theory of heterogeneous processes occurring in the flow using the Langmuir–Hinshelwood–Panchenkov theory.

## 2. Results and Discussion

### 2.1. Influence of the Volumetric rate of Diesel Fuel Supply and Temperature on the Degree of Conversion of Sulfur Compounds

In industry, the process of hydrodesulfurization in the transformation of hydrocarbons of four classes, sulfur, and nitrogen compounds was carried out on installations with a fixed catalyst layer and with a fluidized catalyst layer. The process with a fixed catalyst layer was carried out in the mode of ideal plug flow of the reaction mixture through the catalyst layer.

The results of the influence of the volumetric feed rate on the conversion of diesel fuel in the reaction of hydrodesulfurization of the fraction at different temperatures and constant pressure are shown in Table 1.

**Table 1.** Effect of the hour space velocity of the diesel fraction in the reactor on the degree of conversion of sulfur compounds $x$ at $p = 3.5$ MPa and different temperatures (k—reaction rate constant).

| Space Velocity, $\upsilon$, cm$^3$/(cm$^3$·h) | Temperature | | | | | | | |
|---|---|---|---|---|---|---|---|---|
| | 300 °C | | 340 °C | | 380 °C | | 400 °C | |
| | $x$ | $k$, h$^{-1}$ | $x$ | $k$, h$^{-1}$ | $x$ | $k$, h$^{-1}$ | $x$ | $k$, h$^{-1}$ |
| 1.20 | 0.74 | 1.62 | 0.84 | 2.20 | 0.94 | 3.38 | 0.98 | 4.69 |
| 1.42 | 0.68 | 1.62 | 0.89 | 3.13 | 0.87 | 2.90 | 0.95 | 4.25 |
| 2.02 | 0.56 | 1.66 | 0.78 | 3.06 | 0.78 | 3.06 | 0.90 | 4.65 |
| 3.00 | 0.46 | 1.85 | 0.60 | 2.75 | 0.72 | 3.82 | 0.85 | 5.69 |
| 4.20 | 0.31 | 1.56 | 0.51 | 3.00 | 0.58 | 3.64 | 0.67 | 4.66 |
| Mean | - | 1.660 | - | 2.827 | - | 3.359 | - | 4.789 |
| Standard deviation | - | 0.11 | - | 0.36 | - | 0.39 | - | 0.54 |

In accordance with the generalized quantum-chemical principle, an increase in the temperature of the reaction mixture in the presence of a catalyst increases the number of excited molecules of sulfur compounds RSH + $E$ = RSH *, due to the transition of electrons from HOMO to LUMO in C-S bonds, thereby increasing the rate of the catalytic reaction of the C-S transformation in the H$_2$ stream.

From the data given in Table 1, the following can be noted:

-   with an increase in the volumetric feed rate of the diesel fraction into the reactor, the contact time of the raw material with the catalyst decreases, and the degree of conversion of sulfur compounds decreases at all temperatures;
-   as the temperature increases, the kinetic energy of the movement of molecules increases, and the reserve of potential energy that is consumed in the collision of molecules to excite them increases;
-   polar molecules of sulfur compounds are held by the electromagnetic field of the tetrahedra [MoO$_4$], [NiO$_4$], and [AlO$_4$] for a sufficiently long time and conditions are created for their complete transformation.

According to Table 1, a graph of the dependence of the conversion of sulfur compounds on the volumetric hour speed of supply of the diesel fraction to the reactor is plotted, which is shown in Figure 1.

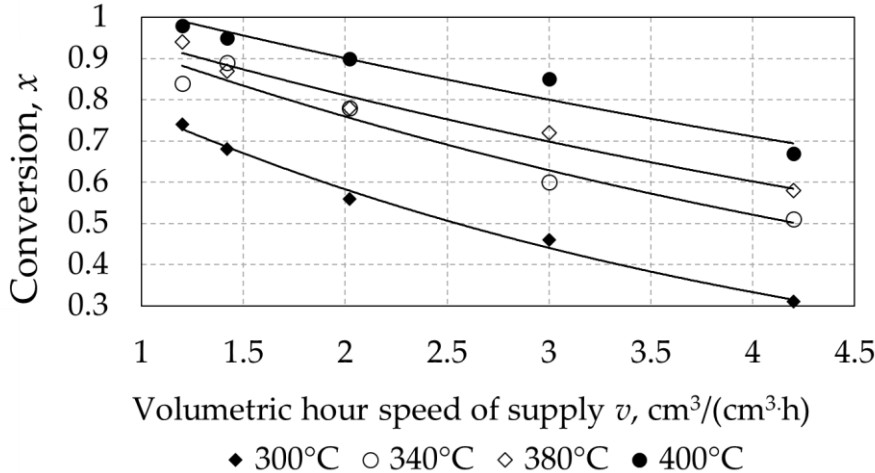

**Figure 1.** Dependence of the conversion of sulfur compounds on the volumetric speed of supply of diesel fraction to the reactor at different temperatures in the reactor, °C.

Figure 1 shows that the conversion of sulfur compounds decreases nonlinearly with an increase in the volumetric feed rate of raw materials into the reactor.

The process of catalytic hydrodesulfurization of the diesel fuel fraction was carried out in a gaseous mixture of raw materials and hydrogen, and the partial pressure of hydrogen was higher than the partial pressure of the diesel fraction. The process was carried out thermodynamically irreversibly, away from the equilibrium state, so the kinetic scheme can be represented in the form of an irreversible (or one-sided) reaction of the 2nd order:

$$RSH + H_2 \rightarrow RH + H_2S \tag{1}$$

In general, the kinetic scheme can be represented as follows:

$$A\ (RSH) + B\ (H_2) \rightarrow C\ (RH) + D\ (H_2S)$$
$$v_0(1 - x) + v_0(\gamma - x) \rightarrow v_0 x + v_0 x \tag{2}$$

where $v_0$ is the volume hour speed of supply of the diesel fuel fraction to the reactor, $x$ is the conversion of sulfur compounds, and $\gamma$ is the mass ratio of hydrogen and raw materials equal to 15:1.

For the presented kinetic scheme, the equations of kinetics are compiled on the basis of the theories of Langmuir, Hinshelwood and Panchenkov.

$$\theta_A = \frac{b_A P_A}{1 + b_A P_A + b_B P_B + b_C P_C + b_D P_D} \tag{3}$$

$$\theta_B = \frac{b_B P_B}{1 + b_A P_A + b_B P_B + b_C P_C + b_D P_D} \tag{4}$$

where $\theta_i$—the fractions of the catalyst surface occupied by the $i$ reactant molecules, $b_i$—the adsorption coefficient of the $i$ reactant, and $P_i$—the partial pressure of the $i$ substance in the gas phase.

The Hinshelwood reaction rate is proportional to the fraction of the catalyst surface occupied by the reactant molecules:

$$w = k \times \theta_A \times \theta_B \tag{5}$$

The reaction rate for a process occurring in the ideal plug flow of the reaction mixture through the catalyst layer is described by the Panchenkov equation:

$$\frac{\upsilon_0 dx}{Sdl} = w \tag{6}$$

$S$, $l$—the surface area of the catalyst, the length of the catalyst layer in the reactor, respectively, $\upsilon_0$—volumetric speed supply of diesel fuel through catalyst layer.

Combining Equations (3)–(6) for a kinetically irreversible hydrodesulfurization reaction, we obtain the following expression in differential form:

$$\frac{\upsilon_0 dx}{Sdl} = k\frac{b_A P_A b_B P_B}{(1 + b_A P_A + b_B P_B + b_C P_C + b_D P_D)^2} \tag{7}$$

Substituting partial pressures for the reagents in Equation (7) and converting the equations, we obtain the following differential equation of kinetics, assuming that the content and partial pressure of hydrogen in the mixture is constant, and the adsorption of reagents at elevated temperatures is weak, i.e., $\Sigma\, b_i P_i \ll 1$:

$$\frac{\upsilon_0 dx}{sdl} = k_1(1 - x) \tag{8}$$

where $k_1 = \frac{k b_A b_B P}{\gamma}$.

As a result of integrating the kinetic Equation (8) in the range from 0 to $x$ and from 0 to l, the following equation is obtained:

$$-\upsilon_0 \ln(1 - x) = k_2 \tag{9}$$

According to Equation (9) and experimental data, the rate constants of the hydrodesulfurization process of the diesel fraction are calculated at different temperatures (Table 1). Table 1 data show that the standard deviation of the rate constant of the hydrodesulfurization reaction of the diesel fraction from the average value is 4.3; 12.6; 4.7; and 3.4% for temperatures of 300, 340, 380, and 400 °C, respectively. These deviations are acceptable for creating a mathematical model of the transformation of sulfur-containing compounds in diesel fuel, using the Arrhenius equation.

### 2.2. Dependence of the Hydrodesulfurization Reaction Rate Constant on Temperature

The Arrhenius equation in general form:

$$\ln k = -\frac{E}{RT} + \ln k_0 \tag{10}$$

$E$—experimental activation energy of the diesel fuel fraction hydrodesulfurization process, $k_0$—pre-exponential factor in the Arrhenius equation. The calculation of $E$ and $k_0$ is carried out by the least-squares regression method according to Table 1. The dependence of the logarithm of the rate constant k on the inverse temperature is shown in Figure 2.

The numerical values of the kinetic constants are:

$$E = 31{,}159.21 \text{ J/mol}, k_0 = 1176.501 \text{ h}^{-1}$$

The equation of the mathematical model, taking into account Equation (9) and the Arrhenius equation, can be presented in the following form:

$$-\upsilon_0 \ln(1 - x) = 1176.501 \cdot e^{-31159/RT} \tag{11}$$

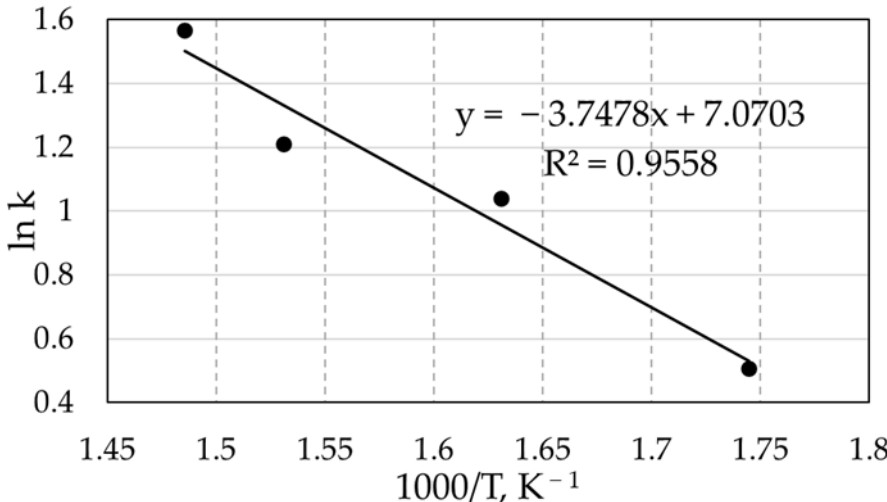

**Figure 2.** Dependence of the logarithm of the rate constant of the hydrodesulfurization reaction of the diesel fraction on the reciprocal absolute temperature ($\ln k = -\frac{3747.8}{T} + 7.0703$, $R^2 = 0.9558$).

The activation enthalpy is equal to:

$\Delta H = E - \mathrm{RT}/2 = 28{,}444$ J/mol, according to the theory of the transition state.

The activation entropy is calculated by the formula:

$$k_0 = \frac{k_B \cdot T}{h} \cdot e \cdot e^{\frac{\Delta S}{R}} \tag{12}$$

$k_B$ Boltzmann constant, $h$—Planck's constant. Logarithm (12), we get the following expression:

$$\ln k_0 = \ln \frac{k_B T}{h} + 1 + \frac{\Delta S}{R} \tag{13}$$

Substituting the numerical values of the parameters in Equation (13), we obtain a change in the entropy of activation:

$$\Delta S = -143 \text{ J}/(\text{mol·K}).$$

According to the transition state theory, the change in the Gibbs energy at 400 °C is equal to:

$$\Delta G * = \Delta H *\text{-}T\Delta S * = 30{,}904 + 673 \cdot 143 = 127{,}143 \text{ J}/\text{mol}$$

The negative value of the change in the entropy of the process reflects the formation of an active catalytic complex based on the interactions of ensembles of tetrahedra and RSH molecules of regular structure with increased binding energy between the molecules of sulfur compounds and catalytic centers. Moreover, the creation of such a complex requires an increased expenditure of energy supplied to the catalyst from the outside (heating of the catalyst and the mixture). The positive value of the Gibbs energy change is defined as the required work (energy) at the highest temperature for the electronic excitation of molecules of sulfur compounds:

$$\Delta G \degree = -W' \tag{14}$$

where $W'$—maximum useful work (without taking into account the loss of part of the work in the form of heat to the environment through the walls of the reactor [37]).

### 2.3. Influence of Pressure in the Hydrodesulfurization Process Reactor on the Conversion of Sulfur Compounds

Increasing the pressure in the reactor increases the number of intermolecular collisions and collisions of reagent molecules with active tetrahedral ensembles on the catalyst surface, which increases the number of adsorbed hydrocarbon molecules, sulfur compounds,

hydrogen, and excited molecules on the active catalyst centers. This leads to an increase in the proportion of the surface occupied by reagent molecules and the rate of chemical reaction according to the theories of Hinshelwood and Langmuir. Excited hydrogen molecules hydrogenate olefin and aromatic hydrocarbon molecules after their excitation, which reduces the formation of coke on the catalyst surface and ensures the high activity of the catalyst surface for a long time of the process. In addition, hydrogen molecules can hydrogenate coke deposits on the inner and outer surfaces of the catalyst, preventing their rapid accumulation. In this paper, depending on the rate of hydrodesulfurization of the diesel fraction, the influence of the pressure of the reaction mixture of the diesel fuel fraction and hydrogen in the reactor on the degree of conversion of sulfur compounds was estimated. Hydrodesulfurization of the sulfur fraction of diesel fuel was carried out at a temperature of 400 °C and a volumetric flow rate of the fraction of $3\ h^{-1}$ for 1 h, the pressure was regulated by a hydrogen bypass valve. The results of the conversion of sulfur compounds in diesel fuel depending on the pressure in the reactor are presented in Table 2.

**Table 2.** Dependence of the sulfur compounds conversion, x, on the pressure P of the reaction mixture in the reactor (υ—volumetric flow rate of the diesel fraction).

| υ, $cm^3/(cm^3\ h)$ | P = 0.55 MPa | | P = 1.10 MPa | | P = 2.2 MPa | | P = 3.5 MPa | |
| --- | --- | --- | --- | --- | --- | --- | --- | --- |
| | $x$ | $-\ln(1-x)$ | $x$ | $-\ln(1-x)$ | $x$ | $-\ln(1-x)$ | $x$ | $-\ln(1-x)$ |
| 1.20 | 0.76 | 1.43 | 0.84 | 1.83 | 0.94 | 2.81 | 0.98 | 3.91 |
| 1.42 | 0.72 | 1.27 | 0.80 | 1.61 | 0.90 | 2.30 | 0.94 | 2.81 |
| 2.02 | 0.60 | 0.92 | 0.78 | 1.51 | 0.88 | 2.12 | 0.90 | 2.30 |
| 3.00 | 0.55 | 0.80 | 0.66 | 1.08 | 0.76 | 1.43 | 0.80 | 1.61 |
| 4.20 | 0.34 | 0.42 | 0.60 | 0.92 | 0.64 | 1.02 | 0.68 | 1.14 |

Table 2 shows that the content of sulfur compounds in the diesel fuel fraction decreases both with a decrease in the volume feed rate of the raw material and with an increase in the pressure of the reaction mixture of hydrogen and the fraction in the reactor. The regularities of changes in the degree of transformation of sulfur compounds from pressure can be described by a parametric equation in an exponential-hyperbolic form:

$$-\ln(1-x) = K\frac{P}{1+P}$$

According to Table 2 and Figure 3, the rate constant of pressure changes in the reactor, *K*, equal to 0.40 is calculated.

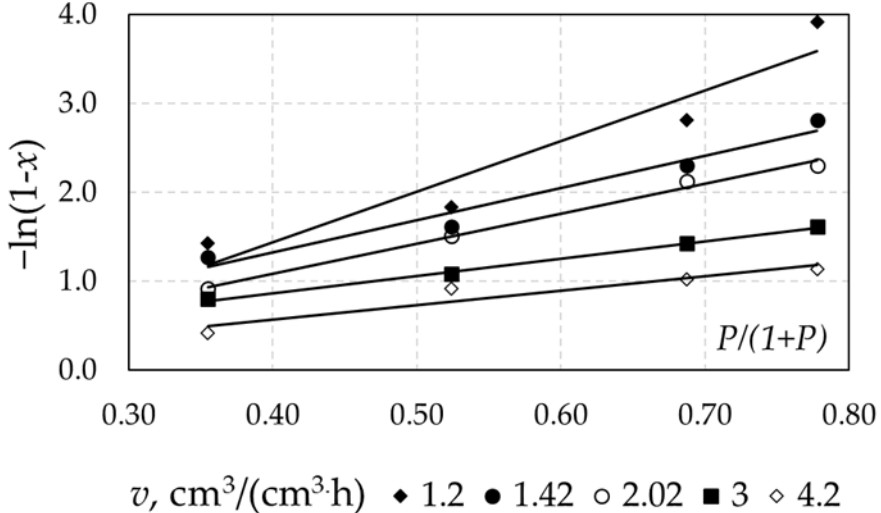

**Figure 3.** Correlation between $-\ln(1-x)$ and $P/(1+P)$ over different volumetric flow rate (υ) of the diesel fraction.

Figure 3 shows that the experimental points fit satisfactorily on a series of straight lines with a constant angle of inclination to the abscissa axis, which makes it possible to determine the average value of the constant and create a parametric model in the form of an expression (15):

$$-\ln(1-x) = 0.40\frac{P}{1+P} \tag{15}$$

According to the experimental data, it is possible to note the presence of some relationship between the volume rate of the fraction supply to the reactor and the pressure in the reactor. The dependence of the degree of conversion ($x$) of diesel fuel on the volumetric rate of diesel fuel supply ($\upsilon$) to the catalyst layer at different pressures is shown in Figure 4.

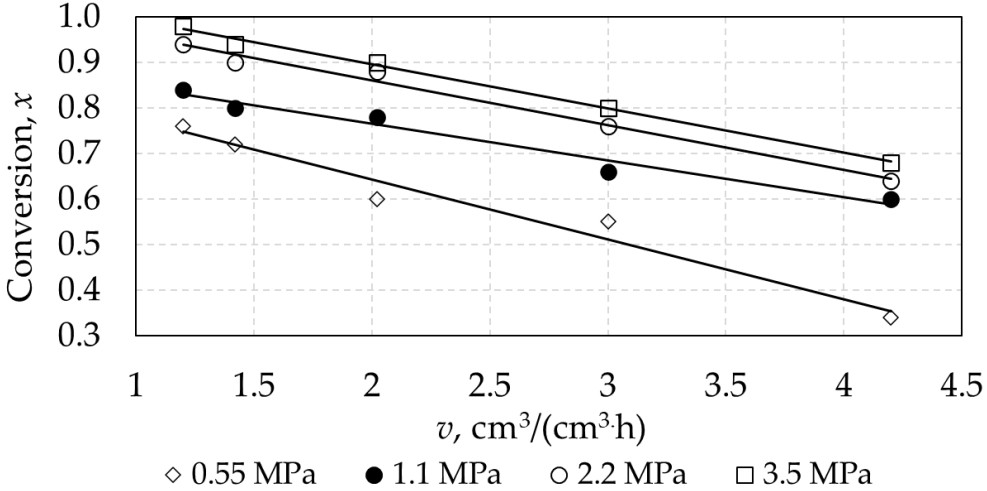

**Figure 4.** Dependence $x = f(\upsilon)$ for different values of $P$.

The parametric equation for this regularity was represented by a complex-exponential equation in the next form:

$$-\ln(1+\upsilon) = \alpha \cdot \ln(1+P) + \beta \tag{16}$$

where $\alpha$ and $\beta$ are constants.

According to Table 2, the anamorphoses of Equation (16) were represented as straight lines (Figure 5).

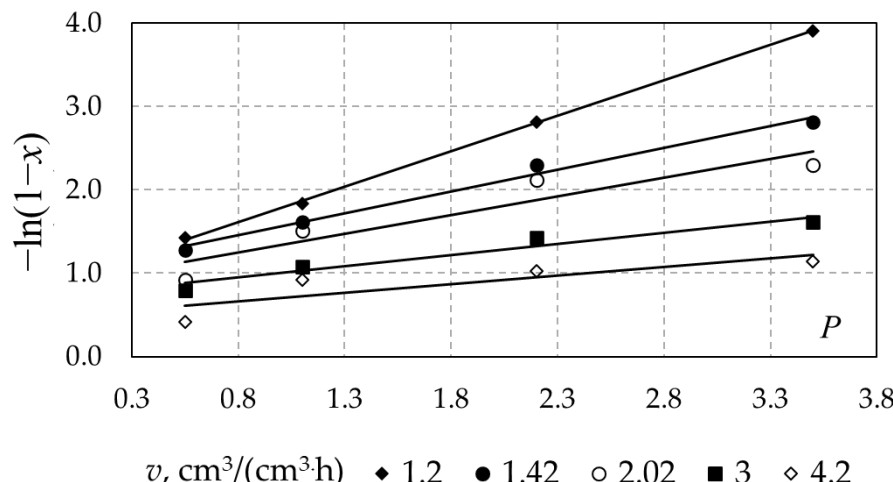

**Figure 5.** Dependence $-\ln(1-x) = f(P)$ for different values of volume rate of fuel supply ($\upsilon$).

Dependence $-\ln(1 - x) = f(P)$ is approximated by a straight line, which indicates the adequacy of this equation to the experimental regularity. The constants of the parametric Equation (16) are $\alpha = 0.806$, $\beta = 0.42$.

The equation of the parametric model was obtained in the following form:

$$\ln(1 + \upsilon) = 0.806 \cdot \ln(1 + P) + 0.42 \tag{17}$$

The parametric models presented in this paper are a new result of the development of a parametric method for analyzing experimental patterns.

### 2.4. Formulation of the Mechanism of Catalytic Hydrodesulfurization of Diesel Fuel

The mechanism of the catalytic hydrodesulfurization of the oil fraction was formulated using the theory of catalysis by polyhedra [27–29] based on data on the acceptor strength of tetrahedra: $[AlO_4]$, $[NiO_4]$, and $[MoO_4]$ that are part of the tetrahedron ensemble $\{AlO_4 \cdot NiO_4 \cdot MoO_4\}$ and taking into account the models in works [38–40]. According to the data of [41,42], taking into account the oxidation state of ions in the composition of tetrahedra, the acceptor strengths of tetrahedra in the catalyst are arranged in a row and are equal to $[MoO_4](0.313) > [AlO_4](0.246) > [NiO_4](0.129)$.

Therefore, in the ensemble of tetrahedra, the highest acceptor strength has $[MoO_4]$-tetrahedron. Then the catalytic ensemble of tetrahedra in the interaction with molecules of sulfur compounds and hydrogen in the mechanism of hydrodesulfurization of the diesel fraction is already an ensemble optimized for the distribution of electrons among the tetrahedra. The hydrodesulfurization process proceeds cyclically and, based on the Generalized quantum mechanical principle, can be represented by a circular scheme (Figure 6) with the letter K denoted for the simplicity of the tetrahedron ensemble, where K = $\{MoO_4\ AlO_4\ NiO_4\}$.

The orbital cycle of the stages of the process of hydrodesulfurization of mercaptane RSH molecules in the presence of an ensemble of tetrahedra is presented below $\{MoO_4 \cdot AlO_4 \cdot NiO_4\}$ with the possible replacement of so me of the oxygen ions by $S^{2-}$.

The mechanism of the catalytic hydrodesulfurization of diesel fuel can be represented based on the Generalized quantum chemical principle formulated in [26–28] as a sequence of the following stages:

I stage. The RSH molecule is diffused from the volume of the reaction mixture to the active ensemble of the tetrahedra K, and under the influence of the electromagnetic field of ions $Mo^{n+}$, $Al^{m+}$, and $Ni^{z+}$ with the symmetry $\Gamma_B$, an electron on the molecular orbital (MO) of the mercaptan molecule with the symmetry of the irreducible representation of the $\Gamma_B$ interacts with the vacant atomic orbital (AO) $nd^0$ of the catalyst active center.

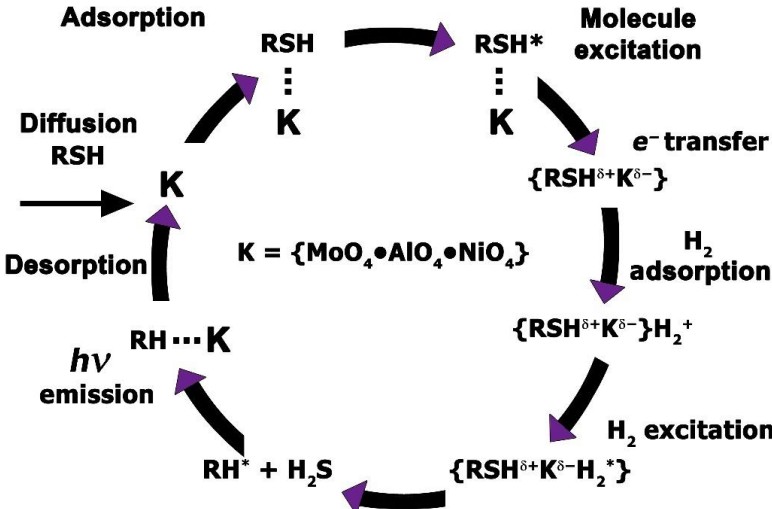

**Figure 6.** Scheme of the mechanism of catalytic hydrodesulfurization of mercaptan molecules (RSH + H2) in the presence of an ensemble of tetrahedra $\{MoO_4 \cdot AlO_4 \cdot NiO_4\}$.

The interaction of the active center K and the molecule of mercaptan ensures the appearance of a full-symmetric representation of ΓА: ΓВ·ΓВ = ΓА. Such interaction is allowed from the point of view of group theory, since the symmetry of the valence $\pi^2_{P_z P_z}$ MO of mercaptan coincides with the $nd^0$-AO symmetry of the active tetrahedron (zero-vacant AO).

A favorable orientation of the flat mercaptan molecule in the electromagnetic field of the tetrahedron with the highest acceptor strength [MoO4]$^{2-}$ occurs. The mercaptan molecule can be oriented vertically or horizontally to the surface of the catalyst.

II stage. The molecule RSH is adsorbed on the active center of the tetrahedron ensemble as a result of the Van der Waals interaction when the mercaptan molecule approaches the catalyst surface. The adsorbing mercaptan molecule contains a pair of free electrons on the sulfur atom and exhibits an increased donor capacity for the [MoO$_4$] tetrahedron, which has an increased acceptor force, stabilized by the [AlO$_4$] and [NiO$_4$] tetrahedra. In addition, the tetrahedra [MoO$_4$] and [AlO$_4$] have an electromagnetic field sufficient to attract a mercaptan molecule (another sulfur-containing compound or unsaturated hydrocarbon) at a distance of $0.8 - 1$ Å. The orientation of RSH molecules on an ensemble of tetrahedra (horizontally or vertically) with its orientation in the field of two tetrahedra is favorable for the excitation of a pair of electrons on the C-S bond.

As was shown above, the adsorption of a mercaptan molecule on active centers in the ground (non-excited) state is also permitted from the position of group theory, since the product of irreducible representations of the vacant atomic orbital of the molybdenum ion and the valence orbital of the mercaptan molecule is equal to the full-symmetric representation (ΓВ·ΓВ = ΓА). This complete symmetry of the complex reflects the possibility of physical adsorption of the molecule on the active centers of the catalyst in the electromagnetic field of polyhedron ensembles.

III stage. Excitation of a physically adsorbed mercaptan molecule on a pair of tetrahedra [MoO$_4$] and [AlO$_4$] with electron transfer from the highest occupied molecular orbital (HOMO) to the lowest unoccupied molecular orbital (LUMO) under the influence of the electromagnetic field of the tetrahedra and under the influence of energy supplied to the complex from outside, of different nature, for example, heat, irradiation.

IV stage. Electron transfer from the HOMO of the mercaptan molecule to the LUMO of the tetrahedron with the formation of an excited mercaptan molecule RSH*.

V stage. Electron transfer from the excited RSH* molecule to the K tetrahedra to form a cation-anion radical $\{RSH^{\delta+}K^{\delta-}\}$.

VI stage. At the same time, the hydrogen molecule H2 is physically adsorbed on the complex, possibly on the neighboring tetrahedron [NiO4], with the formation of the complex $\{RSH^{\delta+}K^{\delta-}\}$H2.

VII stage. Then the excitation of the hydrogen molecule in the complex $\{RSH^{\delta+}K^{\delta-}H_2*\}$ occurs.

VIII stage. In the complex $\{RSH^{\delta+}K^{\delta-}H_2*\}$ charges and bonds are redistributed between the excited hydrogen molecule and the mercaptan molecule according to the law of conservation of bonds and electrons, and paraffin hydrocarbon (RH) and H$_2$S molecules are formed. One of the RH and H$_2$S molecules is in an excited state.

IX stage. The excited molecule emits a quantum of energy and passes into the ground state, in which the valence orbitals have an irreducible representation ΓА, and the vacant atomic orbital of the K centers have an irreducible representation ΓВ. Product ΓА and ΓВ leads to not fully symmetric representation ΓВ = ΓА × ΓВ, and the catalytic complex decays with the release of RH and H$_2$S molecules into the free volume. The cycle is completed; the catalyst is regenerated and enters the next cycle, etc. This is the orbital mechanism of mercaptan hydrodesulfurization.

## 3. Experimental

To study the process of hydrodesulfurization of diesel fuels, an industrial aluminum-nickel-molybdenum oxide catalyst was used. The physicochemical properties of the most common sample are shown in Table 3.

**Table 3.** Physical and chemical properties of Al-Ni-Mo-oxide catalyst.

| Parameter | Quantitative Indicators |
|---|---|
| Bulk density, kg/m$^3$ | 670 |
| Specific surface area, m$^2$/g | 230 |
| Nickel oxide, wt% | 4.0 |
| Molybdenum oxide, wt% | 18.0 |
| Aluminum oxide, wt% | 77.65 |
| Iron oxide, wt% | 0.13 |
| Sodium oxide, wt% | 0.22 |
| The length of the pellets, mm | 6 |
| The diameter of the pellet, mm | 4–5 |
| Color | Light Green |

The composition and properties of the fraction of diesel fuel that was hydrodesulfurized in the experiments are presented in Table 4. The diesel fuel fraction was obtained from the catalytic cracking unit of the Moscow Refinery. The fraction contains a significant amount of aromatic, unsaturated hydrocarbons and sulfur compounds.

**Table 4.** Properties of the investigated diesel fuel.

| Fractional Composition, wt%/Boiling Point of the Fraction, °C | Density $\rho^{20}_4$, g/cm$^3$ | Viscosity $\nu$, sm$^2$/s | Content of Sulfur Compounds, ppm | Concentration, wt% | | | |
|---|---|---|---|---|---|---|---|
| | | | | Aromatic Hydrocarbons | Naphthenes | Saturated Hydrocarbons | Olefins |
| 10/184   50/235                90/321 | 0.9073 | 2.93 | 120 | 18.2 | 26.3 | 41.1 | 14.4 |

The hydrodesulfurization process was carried out on a flow unit with a fixed catalyst layer under the pressure of hydrogen supplied from the cylinder to the reactor through a control valve at temperatures in the catalyst layer of 300, 340, 380, and 400 °C and volume feed rates of 1.2; 1.42; 2.02; 3.0; and 4.2 cm$^3$/(cm$^3$·h), at $P$ = 3.5 MPa. The ratio of hydrogen to raw materials was maintained at 15:1. The analysis of hydrogenated feed was performed by spectrometry on the device SPECTROSCAN (Russian Federation).

## 4. Conclusions

Catalytic hydrodesulfurization of diesel fuel is one of the main processes for the production of commercial diesel fuel of the Euro-5 standard with a residual content of sulfur of no more than 10 ppm and the Euro-6 standard with a content of sulfur <10 ppm.

The process of catalytic desulfurization of diesel fuel has been studied in a wide range of conditions under which environmentally friendly diesel fuel is produced. The studied regularities of the processes of hydrodesulfurization of diesel fuel at different temperatures, volume feed rate of raw materials, and pressure by kinetic and parametric methods are presented.

The regularities of changes in the conversion of sulfur compounds depending on the volume feed rate of raw materials are found.

Based on the theory of Langmuir, Hinchelwood и Panchenkov, the equation of kinetics and a kinetic model that adequately describes the experimental regularity are obtained. The influence of pressure P in the range from 0.55 to 3.5 MPa on the degree of purification of diesel fuel ($x$) was established. The parametric method is used to derive the equation of the function $x = f(P)$ at $T = $ const, and, for the first time, a parametric model is obtained that demonstrates the effect of pressure on the efficiency of the process. The found patterns can be used to select the conditions of the process that provide the required conversion of sulfur-containing compounds.

Thermodynamic characteristics of hydrodesulfurization reaction such as change in Gibbs energy, activation entropy ($\Delta S = -143$ J/(mol·K)), and actination enthalpy ($\Delta H = 28,444$ J/mol) are calculated. Activation energy ($E = 31,159.21$ J/mol), pre-exponential factor in the Arrhenius equation, and the pre-exponential multiplier in the expression for the dependence of the reaction rate constant on temperature ($k_0 = 1176.501$ h$^{-1}$ ) are also obtained. These data are new because, in the literature, only thermodynamic data for equilibrium hydrodesulfurization process are presented [43].

The mechanism of the orbital stage process of transformation in the reaction of hydrodesulfurization of mercaptans is formulated, taking into account the theory of polyhedral catalysis, the Generalized quantum chemical principle, and the theory of groups, which provides significant assistance in determining the possibility or impossibility of the corresponding stage of the catalytic process from the product of irreducible representations.

**Author Contributions:** Conceptualization, I.K.; methodology, E.I., P.G. and I.K.; formal analysis, D.P., V.L., E.I. and P.G.; investigation, D.P., V.L. and I.K.; data curation, V.L., E.I., P.G. and I.K.; writing—original draft preparation, D.P., V.L. and I.K.; writing—review and editing, V.L. and D.P.; visualization, D.P.; supervision, I.K. All authors have read and agreed to the published version of the manuscript.

**Funding:** This work was supported by Russian Ministry of Education and Science within the framework of the state task in the field of scientific activity, topic number FSZE-2022-0002.

**Conflicts of Interest:** The authors declare no conflict of interest.

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
