# Peer review of "Energy Basics of Catalytic Hydrodesulfurization of Diesel Fuels"

_catalysts, doi:10.3390/catal12111301_

Round 1
Reviewer 1 Report
The manuscript entitled “Energy basics of catalytic hydrodesulfurization of diesel fuels” by Petrova is devoted to the study of hydrodesulfurization of diesel fuel containing 120 ppm of sulfur compounds in the presence of an Al-Ni-Mo-O catalyst with a 98% diesel fuel purification rate. The study seems novel and well-organised. But there are several issues that needs significant consideration:
1. Currently, Euro 5 (30 ppm sulfur content) and Euro 6 (10 ppm sulfur content) diesel motor fuels are produced worldwide. Please check the sulfur content in the different Euro diesels and correct it. Seek guidance from this reputed publication of world’s leading journal and also consider referring it. https://doi.org/10.1080/01614940.2020.1780824
2. Please correct the x-axis title of Figure 1. Dependence of the conversion of sulfur compounds on the volumetric speed of supply of diesel fraction to the reactor at different temperatures in the reactor, °C: 1 – 300, 2 – 340, 3 – 380, 4 - 400.
3. Please improve the font of Figure 6. Scheme of the mechanism of catalytic hydrodesulfurization of mercaptan molecules (RSH+H2) in the presence of an ensemble of tetrahedra {MoO4·AlO4·NiO4} to make it less prominent.
4. References section requires significant improvement. DOI is provided with few references, while others are without DOI. Moreover, please make it align with the GFA.
Author Response
Dear Reviewer,
Thanks for your contribution.
Please find our reply to the review below.
1. Sulfur content in Euro-5 and Euro-6 diesel motor fuels is corrected.
Recommended reference is added to the list of references (reference 24 in the article):
Shafiq, I., Shafique, S., Akhter, P., Yang, W., & Hussain, M. (2020). Recent developments in alumina supported hydrodesulfurization catalysts for the production of sulfur-free refinery products: A technical review. Catalysis Reviews, 1–86. doi:10.1080/01614940.2020.1780824
- The x-axis title of Figure 1 is corrected.
- The font of Figure 6 is improved.
- Major references are provided with DOI
Reviewer 2 Report
Main question addressed by the research: The work addresses the energy analysis of catalytic hydrodesulfurization of diesel fuels. Is this really a suitable title? Authors are checking reaction kinetics and performance rather than energy.
Originality and relevance of the topic: The topic is relevant to the field and it considers a potential research gap and it is suitable for the audience in diesel fuels.
Added value of the paper: The manuscript takes into account the study of the thermodynamics, kinetics and mechanism of the hydrodesulfurization of diesel fuels.
Quality of tables and tables: Good but they should be checked for correct symbols and also the captions should include the names of the variables and not the symbols only.
English is good but typos and formatting should be amended.
Specific improvements for the paper to be considered:
- Abstract is too short and general. It should summarize the main findings and applications of the paper.No final values are included for all the parameters in the study.
- Values obtained for all the analysis should be compared with literature. For example enthalpy and entropy obtained should be discussed critically in section 3.2.
- The selection of the optimal conditions is unclear, please add more discussion for all thepossible aspects studied.
- The conclusions are poor and they would need more elaboration so they clearly match the results.
Author Response
Dear Reviewer,
Thanks for your contribution.
Please find our reply to the review below.
- Reaction kinetics and thermodynamics need calculation of different energy changes such as Gibbs energy, enthalpy and activation energy of the reaction. Because of this, the title of the article is suitable.
- Tables are checked and symbols are corrected. The captions with the names of the variables are inserted.
- Typos and formatting are amended.
- Specific improvements for the paper:
- Abstract reflects the main points of the conducted research. The main findings are presented in the conclusion part of the paper, and the final values for the parameters are also included in the conclusion of the study.
- Values obtained for the activation enthalpy and activation entropy couldn’t be compared with the literature data, because the existing literature data are concerned with the equilibrium state of the process but not the activation process and transition state. The following article can be considered as an example of data available: doi:10.1080/10916460500526973
- The searching for the optimal conditions of the process wasn’t the main purpose of the work. But the patterns found can be used to select the conditions of the process that provide the required conversion of sulfur-containing compounds.
- Conclusions are clarified and supplemented.
Round 2
Reviewer 2 Report
Formatting issues should be amended in affiliations, Figure 1 and Table 2.
Author Response
The formatting of Figure 1 and Table 2 has been corrected and is presented as additional information to the corrected article.
